# Characterization of Volatile Substances in Pu-erh Tea (Raw Tea) at Different Storage Times

**DOI:** 10.3390/foods14050840

**Published:** 2025-02-28

**Authors:** Yang Liu, Zhixia Wang, Xinyi Zhang, Hongyu Chen, Dianrong Ruan, Silei Bai, Jianan Huang, Zhonghua Liu

**Affiliations:** 1Key Laboratory of Tea Science of Ministry of Education, Hunan Agricultural University, Changsha 410128, China; lliuyang0630@126.com (Y.L.); wangsophia12@126.com (Z.W.); 17873549721@163.com (X.Z.); chenhongyu9805@163.com (H.C.); silei_bai@hunau.edu.cn (S.B.); 2Yunnan Six Tea Mountain Tea Industry Co., Ltd., Kunming 650000, China; liudachashan@163.com; 3National Research Center of Engineering and Technology for Utilization of Botanical Functional Ingredients, Hunan Agricultural University, Changsha 410128, China; 4Co-Innovation Center of Education Ministry for Utilization of Botanical Functional Ingredients, Hunan Agricultural University, Changsha 410128, China; 5Key Laboratory for Evaluation and Utilization of Gene Resources of Horticultural Crops, Ministry of Agriculture and Rural Affairs of China, Hunan Agricultural University, Changsha 410128, China

**Keywords:** Pu-erh tea, fruity/floral aroma, smoky aroma, storage time

## Abstract

There is a lack of theoretical evidence regarding the transformation of the aroma of Pu-erh tea (raw tea) during long-term storage. In this study, we comprehensively investigate the aroma characteristics of Pu-erh tea (raw tea) from the same manufacturer, stored for different storage times (7–21 years). Sensory evaluation and qualitative and quantitative analysis of volatile substances were performed on the experimental samples. The results showed that the aroma of Pu-erh tea (raw tea) changed from fruity/floral to smoky and fragrance during the storage process. A total of 290 volatiles were identified by HS-SPME/GC×GC-Q-TOF-MS. The key substances for the fruity/floral aroma are fenchene, (E)-1,2,3-trimethyl-4-propenyl-Naphthalene, (+/−-theaspirane, and decanal, and the key substances for the smoky aroma were 2-ethyl-Furan, camphene, 1-methyl-4-(1-methylethenyl)-Benzene, and cis-*β*-Ocimene. The key aroma substances for the fragrance aroma are 1-methyl-4-(1-methylethylidene)-Cyclohexene, *α*-Terpinene, trans-*β*-Ocimene, (E,E)-2,4-Heptadienal, octanal, 2,5-Dimethoxyethylbenzene, 2,4-Dimethylanisole, 1,2,3-Trimethoxybenzene, and 3,4-Dimethoxytoluene. This study helps us to understand further the aroma changes of Pu-erh tea (raw tea) during long-term storage.

## 1. Introduction

Pu-erh tea is a unique geographical indication product from Yunnan, valued by consumers for its unique flavor characteristics and exceptional health benefits [1,2]. It is made from the fresh leaves of Yunnan big-leaf tea trees (*Camellia sinensis* var. *assamica* (*Mast*.) *Kitamura*) grown within the scope of the Protected Geographical Indication area [3]. Based on processing technology, Pu-erh tea is divided into two types: Pu-erh tea (raw tea) and Pu-erh tea (ripe tea). Pu-erh tea (ripe tea) undergoes a unique fermentation process, resulting in a bright red color and a richer mellow flavor [4]. Unlike Pu-erh tea (ripe tea), Pu-erh tea (raw tea) leaves go through the processes such as spreading, killing, twisting, drying in sunlight, refining and steam molding, etc. The resulting Pu-erh tea (raw tea) is yellow and bright, with a strong taste. In the Pu-erh tea market, it is often said that “the older the better”, meaning that Pu-erh tea with a longer storage time has a higher market value than newer tea. During storage, the changes in Pu-erh tea (raw tea) are more pronounced compared to Pu-erh tea (ripe tea) which has undergone fermentation. After storage, Pu-erh tea (raw tea) exhibits a deeper color, a more pronounced flavor, and a richer, long-lasting aroma [5]. The transformations in Pu-erh tea (raw tea) are characterized by a high degree of unpredictability and complexity, making the study of its quality changes during the storage process of significant importance.

Aroma is one of the most important characteristics in evaluating the quality of tea, and studies have shown that its contribution to the quality of tea reaches 25~40%. It can be used as an important indicator for judging the advantages and disadvantages of tea quality characteristics [6,7]. Generally, the characteristics of aroma in various dark teas are unique. Tibetan tea shows an oily aroma, Pu-erh tea emphasizes a stale aroma, and Liupao tea is characterized by stale, woody, and betelnut aromas [8]. Prolonged storage leads to significant changes in the aroma of the tea. The aroma characteristics of compressed white tea after storage evolve from initially sweet, fruity, and floral notes to more pronounced stale, woody, and herbal characteristics [9]. Related studies showed that the storage process has the most significant effect on the aroma of Pu-erh tea (raw tea); after a long time of storage, Pu-erh tea (raw tea) can be transformed into aged tea, and the longer the time, the better the quality [10,11]. Compared with Pu-erh tea (ripe tea), the aroma of Pu-erh tea (raw tea) contains a variety of complex, volatile components, mainly fruity/floral aroma [12]. The characteristic volatiles in Pu-erh tea (raw tea) are linalool, tridecane, caffeine, dihydroactinidiolactone, *β*-violetone, 6,10,14-trimethyl-2-pentadecanone, and dodecane, etc. With long-term aging, Pu-erh tea (raw tea) undergoes complex chemical changes, resulting in a decrease in the levels of low-boiling-point alcohols and hydrocarbons and an increase in some high-boiling-point acids and methoxyphenolic compounds [13]. Therefore, Pu-erh tea (raw tea) stored for a long period should have the quality characteristics of pure aroma, mellow and sweet. Previous studies on the aroma of Pu-erh tea (raw tea) have focused on processing technology [14], varieties [15,16], grades of raw materials [17], different regions [18,19,20], etc. Changes in volatile components related to the aroma of Pu-erh tea during long-term scientific storage have not yet been fully revealed.

With the improvement in testing instruments and methods, a variety of methods are available for the determination of aroma, and GC-MS, which is capable of qualitatively and quantitatively characterizing volatile compounds, is the most frequently used analytical technique for the detection and analysis of aroma compounds in Pu-erh tea [21]. The peak capacity of GC-MS is insufficient, and the separation capacity is limited and the sensitivity is low, so it can only identify a limited number of separable compounds [22]. The method of GC×GC-Q-TOF-MS is a new analytical technique with higher sensitivity and resolution for the determination of aroma components. Compared with GC-MS, GC×GC-Q-TOF-MS is an emerging analytical technique with higher sensitivity, peak capacity, and resolution for the detection of aroma components. The combination of gas chromatography and high-resolution quadrupole time-of-flight mass spectrometry (Q-TOF-MS) has been proven to be applicable in various analytical fields, including flavor research [23] and volatile analysis [24,25], and has been demonstrated to be a powerful analytical tool. Therefore, in this study, GC×GC-Q-TOF-MS combined with headspace solid-phase microextraction (HS-SPME) was used to analyze the volatile aroma components of Pu-erh tea (raw tea) processed from the same origin with different storage times. The results revealed the changing rules of sensory quality and aroma composition of Pu-erh tea (raw tea) with different storage times, with a view to providing certain theoretical support for the storage conditions of Pu-erh tea (raw tea).

## 2. Materials and Methods

### 2.1. Materials and Reagents

The 14 Pu-erh tea (raw tea) samples were selected from Yunnan Six Tea Mountain Tea Industry Co. Ltd., stored in Yunnan, China. The sample information is shown in Table 1. All samples are made from the same raw materials, using the same processing technology, and stored under the same conditions every year. The storage conditions are a dry, ventilated, and odorless environment at a temperature of 20~30 °C and relative humidity of 40%~50% for a long time. The tea samples were evenly sampled according to the quadrature method and stored in the refrigerator at 4 °C.

### 2.2. Sensory Evaluation

Five trained group members (three men and two women) from Hunan Agricultural University were selected to conduct sensory evaluations of tea samples. Prior to the formal experiments, each assessor underwent at least 90 h of specialized training to ensure they could accurately describe the sensory characteristics of tea. During the evaluation, the assessor was required to evaluate the appearance, soup color, aroma, taste, and leaf base of each sample and to provide the key terms to describe quality characteristics collectively according to the Methods of Sensory Review of Tea (GB/T 23776-2018 [26]) and Geographical Indication Products pu-erh tea (GB/T 22111-2008 [3]). A total of 3 g of sample was placed in a special cylindrical evaluation cup, added with 150 mL boiling water, covered, soaked for 5 min, and strained into the evaluation bowl to evaluate. Each assessor was evaluated independently without any discussion. Each sample was repeated three times. After each evaluation, the next sample was evaluated after a 10 min rest. Evaluation environment requirements were that it was clean and odorless, with a temperature of 20–25 °C. The average value of the evaluation results of the five evaluators was taken as the final result of the sample.

### 2.3. HS-SPME/GC×GC-Q-TOF-MS Conditions

HS-SPME Extraction Conditions: First, 0.5 g of tea sample powder was weighed and placed into a 15 mL headspace vial (Agilent, Santa Clara, CA, USA). Then, a magnetic force rotor (6 × 8 mm, Anpel, Shanghai, China), boiled water (5 mL), and 10 µL of ethyl caprate (10 mg/L, Sigma-Aldrich, St. Louis, MO, USA) was added to the headspace vial; the headspace vial cap was tightened immediately and it was placed on a thermostatic magnetic stirring heating table (80 °C, 400 rpm, Talboys, NJ, USA) for 10 min. After that, the 50/30 μm divinylbenzene/carboxen/polydimethylsiloxane (DVB/CAR/PDMS) fiber (Supelco, Bellefonte, PA, USA) was pushed out to a position 1 cm from the liquid surface and adsorbed for 30 min at 80 °C at an agitation rate of 400 rpm. Finally, the fiber was inserted into the GC×GC-Q-TOF-MS injector for data collection and analysis, and thermal analysis was performed at 250 °C for 5 min.

The volatile compounds were analyzed utilizing GC×GC-Q-TOF-MS, incorporating an Agilent 7250 GC-Q-TOF-MS instrument (Agilent, Santa Clara, CA, USA) equipped with a solid-state thermal modulator (SSM 1820, J&X Technologies, Shanghai, China).

Gag Chromatographic Conditions: The inlet temperature was 250 °C; the carrier gas was high-purity He_2_ helium (He_2_ ≥ 99.999%) at a flow rate of 1.1 mL/min; the temperature increase program—the initial column temperature was 40 °C, and the temperature was increased to 180 °C at a rate of 4 °C/min, and then the temperature was increased to 250 °C at a rate of 20 °C/min and kept for 1 min for a total of 40.5 min; the shunt injection was performed at a ratio of 10:1.

Mass Spectrometry Conditions: Ion source EI, ionization energy 70 eV, ion source temperature 230 °C; the transmission line temperature is 280 °C; the acquisition range 45–500 *m*/*z*; acquisition rate is 50 Hz/s. The ion source temperature is 230 °C; the transmission line temperature is 280 °C; the acquisition range is 45–500 *m*/*z*, acquisition rate is 50 Hz/s.

### 2.4. Relative Odor Activity Value of Volatile Components of Tea

Relative odor activity value (ROAV) is a method for determining the key flavor compounds of foodstuffs by combining the thresholds of guard compounds, and it is used to elucidate the contribution of each aroma compound to the overall aroma characteristics of the samples. The ROAV is directly proportional to its contribution to the overall aroma. In general, a volatile with ROAV ≥ 1 is regarded as the key volatile affecting the aroma of the tea leaves.(1)$${\mathrm{RO}\mathrm{AV}}_{i}=\frac{C_{i}}{{\mathrm{OT}}_{i}}$$
where C_i_ denotes the content of mass concentration of compound i in water; OT_i_ (Odor Threshold) denotes the aroma threshold of compound i in water.

### 2.5. Qualitative and Quantification of Volatile Compounds

Qualitative analysis: The NIST standard library was referenced to match the detected substances. Then, the final qualitative analysis according to the relative retention time of each volatile substance was conducted.

Quantification analysis: The internal standard method was used to quantify the volatile flavor compounds in the aroma of tea. Ethyl caprate was selected as the internal standard. The calculation formula is as follows:(2)$$\;f=\frac{\frac{A_{s}}{M_{s}}}{\frac{A_{r}}{M_{r}}}$$
where A_s_ and A_r_ are the peak areas or peak heights of the internal standard and control, respectively, and M_s_ and M_r_ are the amounts of the internal standard and control added, respectively. Then, a sample of the component solution containing the internal standard was taken, the chromatogram recorded and the content (M_i_) calculated from the peak response of the component solution containing the internal standard.(3)$$\;M_{i}=\frac{f\;\text{× }A_{i}}{\frac{A_{s}}{M_{s}}}$$
where M_i_ is the analyte concentration in mg/kg, A_i_ and A_s_ are the peak area or peak height of the substance to be measured and the internal standard, respectively, and M_s_ is the amount of internal standard added.

### 2.6. Data Processing

The volatile components of all the samples were analyzed qualitatively by mass spectrometry using Canvas software (version 2.5.0) with the NIST 20 database and were screened by forward match ≥ 700 and reverse match ≥ 800, combined with retention index (RI). Principal component analysis (PCA) and hierarchical clustering analysis (HCA) were performed using Simca14.1 (Umetrics, San Jose, CA, USA). The scatter plots and heatmaps were generated using MetaboAnalyst 6.0 (https://www.metaboanalyst.ca/ (accessed on 1 January 2025) and OmicStudio (https://www.omicstudio.cn/tool, accessed on 1 January 2025), the stacked bar charts were created using GraphPad Prism 10.1.2, and the radar charts were constructed using Excel.

## 3. Results and Analysis

### 3.1. Sensory Evaluation of Pu-erh Tea During Storage

Sensory evaluation results showed that storage time significantly affected the flavor quality of 14 Pu-erh tea (raw tea) samples (Table 2). In the storage process, the soup color of the tea becomes darker, yellow to orange-red, and the taste becomes more mellow. Especially, the aroma characteristics showed regular changes (Figure 1). Among them, the aroma of P1, P2, P4, and P6 is mainly fruity/floral, and the degree of the aroma gradually decreased; the P5, P9, and P10 are mainly smoky aroma, and the degree of the aroma gradually increased; the aroma of P7, P8, P12, and P13 are mainly fragrance, while the aroma of P11 and P14 are also fragrance, but to a stronger extent and purer. The aforementioned transformations suggest that the aroma profile of Pu-erh tea (raw tea) evolves progressively over time from a fruity/floral aroma to a smoky aroma and then to a fragrance aroma. Moreover, the intensity of the aroma also changes accordingly. According to the different aroma types and intensity, we divided P1, P2, P4, and P6 into the first group, P5, P9, and P10 into the second group, P7, P8, P12, and P13 into the third group, and P11 and P14 into the fourth group.

### 3.2. Volatile Substances Were Qualitatively and Quantitatively Determined by HS-SPME/GC×GC-Q-TOF-MS

To analyze the chemical mechanism of storage time on the aroma of Pu-erh tea (raw tea), HS-SPME/GC×GC-Q-TOF-MS was further used for the qualitative and quantitative analysis of volatile components. A total of 290 volatile compounds were identified in 14 Pu-erh tea (raw tea) samples (Appendix A). The volatiles are categorized into 11 major groups according to their functional groups (Figure 2 and Figure 3), including 55 aromatic hydrocarbons, 51 olefins, 36 ketones, 34 alcohols, 32 aldehydes, 24 esters, 24 heterocyclic compounds, 20 ethers, 7 phenols, 6 alkanes, and 1 acid.

Among the volatiles, the highest proportion of the 14 Pu-erh tea (raw tea) samples was alcohols, which accounted for 20.76~34.04% of the total aroma component, followed by olefins and aldehydes, which accounted for 12.98~30.13% and 15.78~22.76% of the total aroma components, respectively. During the storage process, the contents of heterocyclic compounds and aldehydes increased and then decreased, with a sharp increase to 7.14% and 16.25% in the 13th year of storage. In contrast, the levels of olefins increased and then decreased in the opposite direction, with the highest content of 28.03% observed in the 16th year of storage. Overall, the alcohols exhibited a trend of decreasing and then increasing, with the lowest content of 20.76% in the 17th year of storage. Ketones and aromatic hydrocarbons showed an overall decreasing trend. The contents of ester, phenol, ether, acid, and alkane were lower, among which the ester and phenol showed a gentle increase in the storage process. The levels of ethers showed an overall increased trend, and the contents of ethers increased sharply in the 20th year of storage, with a maximum of 11.57%, and the contents of alkanes and acids showed a decreasing trend overall.

### 3.3. PCA Combined with HCA Was Used to Group Samples of Different Years

A total of 290 aroma components from 14 Pu-erh tea (raw tea) samples were analyzed by principal component analysis (PCA) to preliminarily assess the magnitude of differences between the samples. Samples that are close together on the score plot may indicate similarity in metabolic characteristics. The PCA plot (Figure 4A) shows that the samples from 2002 and 2005 (P14 and P11) are distributed in the same region. The samples from 2003, 2004, 2009, and 2010 (P13, P12, P8, and P7) are clustered in another region. The samples from 2011, 2013, 2015, and 2016 (P6, P4, P2, and P1) are grouped in a separate area, and the samples from 2006, 2007, 2012, and 2014 (P10, P9, P5, and P3) are distributed in a distinct region. Samples located in the same region suggest similarity in their volatile components and minimal differences between the samples, allowing them to be grouped into a single category. In addition, the further the component is from the Origin on the PCA loading plot, the greater the contribution to sample classification is considered. As can be seen from Figure 4B, these five substances: trans-(−)-5-methyl-3-(1-methylethenyl)-Cyclohexene, L-α-Terpineol, Limonene, 2-[(2R,5S)-5-ethenyl-5-methyltetrahydrofuran-2-yl]propan-2-ol, o-Cymene may be the key substance affecting the grouping.

To further refine the grouping and similarity of the samples, hierarchical clustering analysis (HCA) was subsequently applied to the 14 Pu-erh tea (raw tea) samples (Figure 4B). The aim of HCA is to classify the samples based on the similarity of their aroma components, thereby uncovering the underlying structure and distinct aroma types among the samples. The HCA dendrogram shows that the 14 samples are divided into 4 groups, and this grouping result is consistent with the PCA results. Combined with the results of sensory evaluation, 14 samples of Pu-erh tea (raw tea) could be divided into 4 groups: the first group A1 (P1, P2, P4, P6), the second group A2 (P3, P5, P9, P10), the third group A3 (P7, P8, P12, P13), and the fourth group A4 (P11, P14).

### 3.4. Differences and Characterization of Volatile Substances in Different Groups

To analyze the key volatile compounds that affect the changes in the aroma profile of Pu-erh tea (raw tea) during storage, the ROAVs of volatile compounds were calculated. ROAV quantitatively represents the intensity of aroma components by combining compound concentration and odor threshold (OT) value [27]. The higher the ROAV value, the stronger the contribution of the compound to the aroma. In general, ROAV ≥ 1 indicates that aroma compounds have a significant effect on the aroma of tea and compounds with ROAV ≥ 10 are generally considered key aroma substances [28]. Table 3 lists 51 volatile compounds with ROAV ≥ 1 in Pu-erh tea (raw tea) during storage. The main contributing substances to the aroma are aldehydes, aromatic hydrocarbons, ketones, and alcohols, followed by heterocyclic compounds, esters, and olefins. Alcohol compounds typically present “floral”, “sweet”, or “woody” scents. Ketones are generally considered the primary contributors to the “woody” aroma of Pu-erh tea. Aldehydes are the main contributors to the “green”, “grassy”, and “fatty” aromas of Pu-erh tea. Most aldehydes are formed by the oxidation of fatty acids in the tea. Among these compounds, the ROAVs of 4-(2,6,6-trimethyl-1-cyclohexen-1-yl)-3-Buten-2-one, (E)-Damascenone, Linalool, and (E)-2-Nonenal were all higher than 100 in all 14 samples. It is concluded that these four aroma components may be the key components of the basic aroma of Pu-erh tea (raw tea). Notably, 28 have a ROAV ≥ 10, indicating that they may play a key role in distinguishing the aroma characteristics of different groups.

To further determine the key aromatic compounds of different groups, the Variable Importance in Projection (VIP) method was applied to quantify the impact of each volatile compound on the classification of the samples. The VIP value effectively reflects the importance of each volatile substance in the model. It is commonly recognized that when the VIP ≥ 1, the volatile substance significantly influences the classification result. In groups A1, A2, A3, and A4, there are 18, 15, 17, and 15 compounds, respectively, with VIP ≥ 1. These compounds make significant contributions to the differentiation of samples into different groups (Figure 5).

As shown in Figure 5, there were 18 key volatiles in Group A1 that satisfied VIP ≥ 1. Among them, linalool, trans-linalool oxide (furanoid), and decanal showed floral and fruity aroma [29,30]; Cedrenol, 2,2,6-trimethyl-Cyclohexanone showed a fruity aroma [31,32,33]; 3,7-dimethyl-1,5,7-Octatrien-3-ol showed a clean and fruity aroma [34,35,36]; 2-Nonen-1-ol showed a green aroma; benzaldehyde showed a nutty, fruity, and burnt aroma [37]; 2-[(2R,5S)-5-ethenyl-5-methyltetrahydrofuran-2-yl]propan-2-ol had a floral aroma [38]; (+/−)-theaspirane had a honey aroma. Combined with the results of the sensory evaluation, Group A1 showed a fruity/floral aroma, so these substances are the key volatile for the formation of the fruity/floral aroma.

There were 15 key volatiles of Group A2 that satisfied VIP ≥ 1. Among them, cis-*β*-Ocimene is floral and fruity [39]; 2-ethyl-Furan has a caramelized aroma; benzaldehyde has a nutty, fruity, and caramelized aroma; limonene, cedrenol, 2,2,6-trimethyl-Cyclohexanone has a fruity aroma; hexanal has a fatty aroma; 3,7-dimethyl-1,5,7-Octatrien-3-ol has a clean and fruity aroma; camphene has a floral aroma; cis-*β*-Ocimene has a floral and fruity aroma; o-Cymene has a smoky and woody aroma; and 1-methyl-4-(1-methylethenyl)-Benzene has a fruity, woody odor [40]. Combined with the results of the sensory evaluation, Group A2 was found to have a smoky aroma, so these substances are the key volatile for the formation of smoke aroma.

There were 17 key volatiles in Group A3 that satisfied VIP ≥ 1. Among them, 2-methyl-Cyclopentanol showed a fresh and fruity aroma; linalool and α-Terpinene showed a fruity/floral aroma; 1-methyl-4-(1-methylethylidene)-Cyclohexene showed a woody aroma; hexanal showed a fatty aroma; limonene, trans-*β*-Ocimene, and *γ*-Terpinene showed a fruity aroma; benzaldehyde showed a nutty, fruity, and caramelized aroma; (E,E)-2,4-Heptadienal, trans-linalool oxide (furan) showed a fresh and fat aroma [41,42]; and o-Cymene showed a woody, smoky aroma. Combined with the results of the sensory evaluation, Group A3 was characterized by a fragrance aroma, so these substances are the key volatile for the formation of a fragrance aroma.

There were 15 key volatiles in Group A4 that satisfied VIP ≥ 1. Among them, 2-Nonen-1-ol and 2-[(2R,5S)-5-ethenyl-5-methyltetrahydrofuran-2-yl]propan-2-ol showed a fresh aroma; octanal showed a fragrance and fruity aroma; benzaldehyde has a nutty, fruity, and smoky aroma. Combined with the results of the sensory evaluation, Group A4 was characterized by a clear aroma, so these substances are the key volatile for the formation of a fragrance aroma.

Through a comprehensive evaluation of these four groups of key volatile compounds, three substances are present in all four groups: trans-(−)-5-methyl-3-(1-methylethenyl)-Cyclohexene, L-*α*-Terpineol, and benzaldehyde. These compounds may play a role in coordinating the floral, fruity, fresh, and smoky aromas of Pu-erh tea (raw tea). During storage, Trans-(−)-5-methyl-3-(1-methylethenyl)-Cyclohexene increased in the 11th year of storage, and its content was relatively stable between the 12th and 15th years of storage. The contents of L-α-Terpineol and benzaldehyde showed an overall increasing trend. The formation of *α*-terpineol is dependent on post-fermented microbial activities [31]. The amino acid-derived volatile benzaldehyde spike may be due to the oxidation of toluene. This key aromatic compound is consistent with previous research findings, and the trend of its changes during storage shows similarity. This further suggests that the variations of these volatile compounds during storage play a crucial role in the transformation of the aroma characteristics [43].

The unique substances in Group A1 were fenchene, (E)-1,2,3-trimethyl-4-propenyl-naphthalene, (+/−)-theaspirane (honey), and decanal (fruity, floral), which were the key substances for the fruity/floral aroma types. Decanal was considered to be the most important aroma component due to its ROAV ≥ 100 in Pu-erh tea, which is consistent with previous conclusions [44]. The unique substances in Group A2 were 2-ethyl-Furan (burnt), camphene (floral), 1-methyl-4-(1-methylethenyl)-Benzene (woody), and cis-*β*-Ocimene (fruity, floral), which are the key aroma-presenting substances for the smoke aroma type. At the same time, all four of these substances have ROAV ≥ 1, so they are the key aroma substances of the smoky aroma type. Compared with Groups A1 and A2, the unique substances in Group A3 were 1-methyl-4-(1-methylethylidene)-Cyclohexene (woody), *α*-Terpinene (fruity, floral), trans-*β*-Ocimene (fruity), *γ*-Terpinene (fruity), (E,E)-2,4-Heptadienal (fragrance), and octanal (fragrance). Among them, 1-methyl-4-(1-methylethylidene)-Cyclohexene, *α*-Terpinene, trans-*β*-Ocimene, (E,E)-2,4-Heptadiena, and octanal have ROAV ≥ 1. This indicates that these are the key aroma-presenting substances for the fragrance aroma type. Octanol was also identified in Group A4, and its ROAV value was greater than 100. This indicates that octanol is a key component in the formation of aroma types and plays a significant role in influencing the richness of the aroma. MA’s research shows that octanal is considered the characteristic active aroma compound of Pu-erh tea with a green flavor [45]. The unique substances of Group A4 were 2,5-Dimethoxyethylbenzene, 2,4-Dimethylanisole, 1,2,3-Trimethoxybenzene, 3,4-dimethoxytoluene, which interacted with other substances to affect the persistence and richness of the aroma. O-methylation promotes simple phenols into methoxyphenols, which contribute to fragrance changes during storage [46]. In the samples of Group A4, the contents of alcohols and methyl groups were higher than those of other samples, especially in the 21st year of storage; the methyl groups increased sharply. The methyl group is the key component of the aging flavor [47,48,49,50], which indicates that the samples are in the process of transformation from fresh flavor to aged flavor. With further storage, it was hypothesized that the aroma of the samples should be gradually transformed into an aging aroma.

### 3.5. Key Substances of Aroma Quality of Stored Aging Pu-erh Tea (Raw Tea)

Groups A3 and A4 were both characterized by fragrance aroma, but the aroma of Group A4 was stronger than that of Group A3. Groups A3 and A4 both contained octanal, which is a key volatile substance in the fragrance aroma of Pu-erh tea (raw tea). Compared with Group A3, Group A4 contained 2,5-Dimethoxyethylbenzene, 2,4-Dimethylanisole, 1,2,3-Trimethoxybenzene, and 3,4-Dimethoxytoluene, which are the key substances for the aging aroma. This indicates that the samples are in the process of transformation into an aging aroma. Pang’s research found that these methyl compounds are key volatile substances in Pu-erh tea (raw tea) [12]. However, the aging aroma of Group A4 was not revealed, which might be due to the masking effect of some substances on these four methyl groups, affecting the presentation of the aging aroma. The correlation analysis of the substances in Groups A3 and A4 (Figure 6) showed that the substances significantly negatively correlated with 2,5-Dimethoxyethylbenzene, 2,4-Dimethylanisole, 1,2,3-trimethoxybenzene, and 3,4-Dimethoxytoluene were 1-ethyl-1H-Pyrrole-2-carboxaldehyde and 1-Hexanol. These two substances have a masking effect on methyl compounds. Substances significantly and positively correlated with 1-ethyl-1H-Pyrrole-2-carboxaldehyde and 1-Hexanol was benzyl alcohol. The substance synergistically enhanced 1-ethyl-1H-Pyrrole-2-carboxaldehyde and 1-Hexanol. As a result, both A3 and A4 groups showed a fragrance aroma, but the overall aroma of Group A4 was richer and long-lasting. This also indicates that Group A4 has a richer variety of substances. By comparing the substances in Groups A3 and A4, it was found that Group A4 had 19 more substances than Group A3. They are mainly concentrated into six categories: ethers, esters, olefins, aromatic hydrocarbons, phenols, and heterocyclic compounds. Among them, ethers are mainly methoxylated substances, which enhance the persistence of aroma. At the same time, aldehydes and esters of floral and fresh aroma inhibit the stale odor of methoxylates and improve the overall acceptability of the aroma [51]. As a result, the aroma of Group A4 was clear and rich, with better persistence.

### 3.6. Discussion and Analysis

The 14 samples can be categorized into fruity/floral aroma, smoky aroma, fragrance aroma, and stronger fragrance aroma based on fragrance type and intensity. The results of the evaluation group are completely consistent with the PCA and HCA grouping results.

In the four subgroups, the production years of the samples were not consecutive, and it was hypothesized that there might be differences in storage conditions. The different production years and storage bins resulted in differences in storage factors such as moisture, room temperature, and oxygenation. Since this batch of samples is from the same manufacturer with unified production and unified storage, storage reasons do not do too much analysis. It is also possible that precipitation, light, and temperature conditions vary from year to year, resulting in changes in the environment of the tea plant in the same place of origin. Ahmed [52] showed that an increase in precipitation resulted in a 50% decrease in the content of catechins and total alkaloidal purines, which are functional components of tea, and an increase in the concentration of total polyphenolic substances and antioxidant activity. Liu et al. [53] found that drought affected the water status as well as physiological and metabolic processes in tea, resulting in lower yields and poorer bud quality. Ahme and Bhattacharya [54,55] demonstrated that drought stress also increased phenolic compounds and antioxidant activity in volatile metabolites of tea trees, thus affecting the aroma formation of tea leaves. The environment of tea plants may vary, and the quality of tea leaves may vary from year to year due to the influence of the environment, so the samples from non-consecutive years of production were assigned to the same group. These key aromatic substances may be used as one of the indicators to predict the aging of Pu-erh tea (raw tea). The samples in this study are still in the process of aging, and the transformation mechanism is still unclear. Since the maximum storage period of the samples in this study was 21 years, it is still worthwhile to investigate the changes in the long-term storage.

## 4. Conclusions

In this study, 14 samples were categorized into fruity/floral aroma, smoky aroma, and fragrance aroma according to different aroma types, and the key components of each aroma type were identified. Fenchene, (E)-1,2,3-trimethyl-4-propenyl-Naphthalene, (+/−)-theaspirane, and decanal were identified as the key components of the fruity/floral aroma. 2-ethyl-Furan, camphene, 1-methyl-4-(1-methylethenyl)-Benzene, and cis-*β*-Ocimene were identified as the key components of the smoky aroma. The key aroma substances for the fragrance aroma are 1-methyl-4-(1-methylethylidene)-Cyclohexene, *α*-Terpinene, trans-*β*-Ocimene, (E,E)-2,4-Heptadienal, octanal, 2,5-Dimethoxyethylbenzene, 2,4-Dimethylanisole, 1,2,3-Trimethoxybenzene, and 3,4-Dimethoxytoluene. These substances can be used as distinguishing substances in the storage stage of Pu-erh tea (raw tea), and the flavor presented can also be used as one of the factors in whether the storage conditions are suitable.

The results of the study showed that the aroma of Pu-erh tea (raw tea) was gradually transformed from fruity/floral to smoky and fragrance aroma, and finally to an aging aroma, which is the change rule of the aroma of Pu-erh tea (raw tea). The biggest difference between the 14 samples of Pu-erh tea (raw tea) is whether the volatile components contain methyl substances or not. With the increase in storage year, the content of methyl substances such as 3,4-Dimethoxytoluene and 1,2,3-Trimethoxybenzene in the tea samples increased gradually. Methyl substances are the key components of the aging aroma, and the aroma of the samples should be gradually transformed into the aging aroma with continued storage. At present, the reason for the formation of methyl groups is ambiguous, and further research is needed.

## Figures and Tables

**Figure 1 foods-14-00840-f001:**
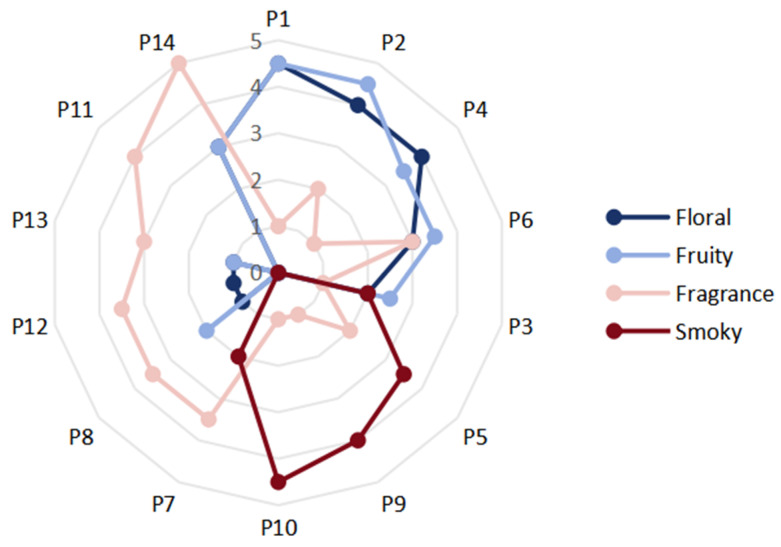
Aroma evaluation radar chart.

**Figure 2 foods-14-00840-f002:**
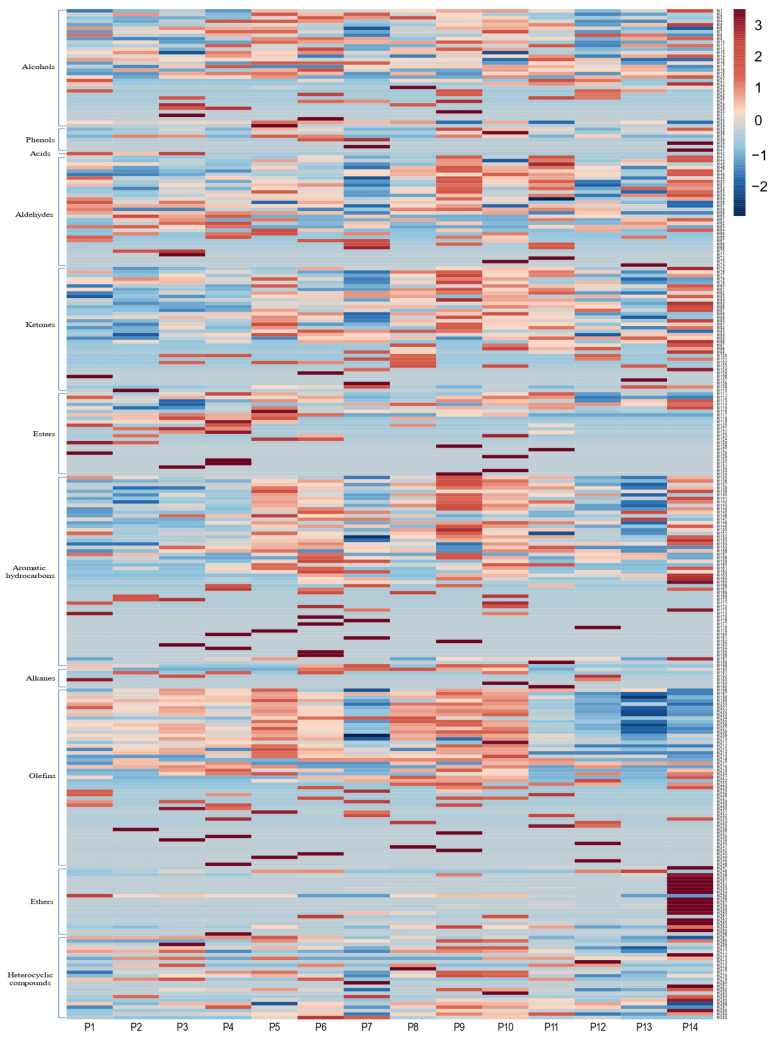
Heat map of 290 volatile components.

**Figure 3 foods-14-00840-f003:**
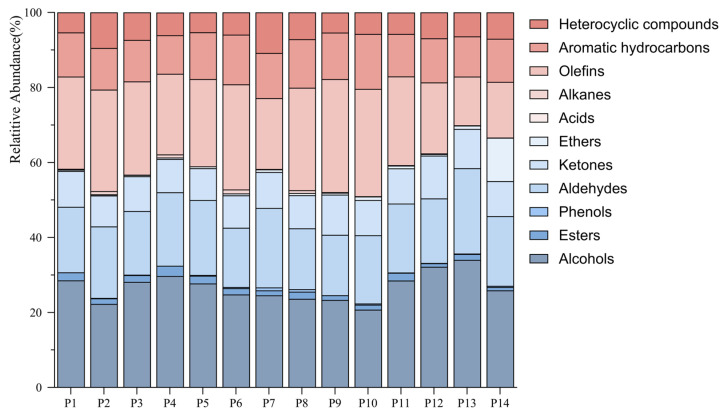
Classification and percentage of volatile compounds.

**Figure 4 foods-14-00840-f004:**
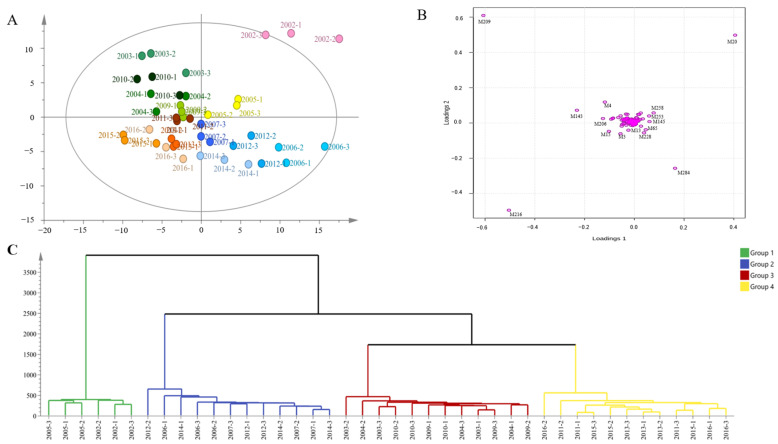
(**A**): Plot of principal component analysis; (**B**): Loading plot of principal component analysis; (**C**): Hierarchical cluster analysis (HCA).

**Figure 5 foods-14-00840-f005:**
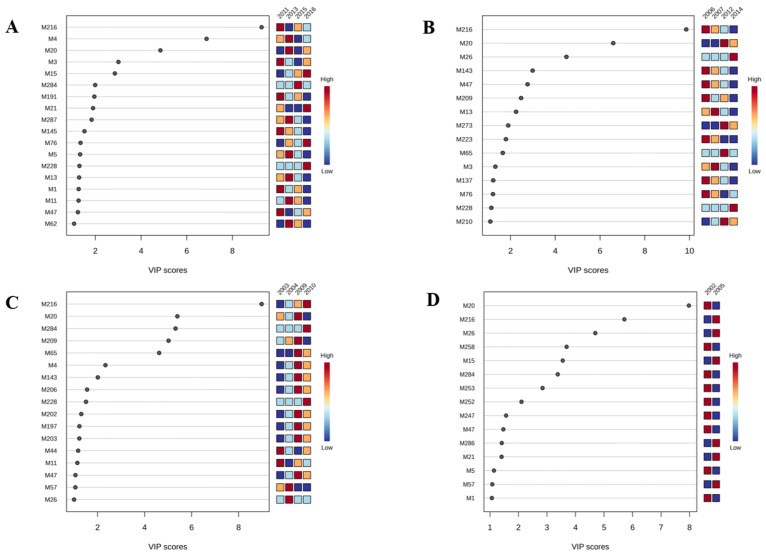
(**A**): VIP score of volatile substances in group A1 (VIP ≥ 1, *p* < 0.05); (**B**): VIP score of volatile substances in group A2 (VIP ≥ 1, *p* < 0.05); (**C**): VIP score of volatile substances in group A3 (VIP ≥ 1, *p* < 0.05); (**D**): VIP score of volatile substances in group A4. (VIP ≥ 1, *p* < 0.05).

**Figure 6 foods-14-00840-f006:**
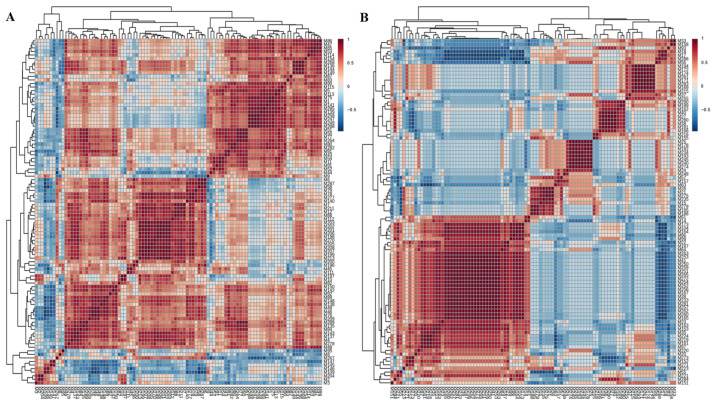
Group A3 and Group A4 correlation analysis heat map ((**A**): Common substances of Groups A3 and A4, (**B**): Non-common substances in Groups A3 and A4).

**Table 1 foods-14-00840-t001:** Information on Pu-erh tea (raw tea) samples.

No.	Sample Name	Year of Manufacture	Storage Time (Years)	Sample Number
1	Liushan Pu-erh tea (raw tea)	2016	7	P1
2	Liushan Pu-erh tea (raw tea)	2015	8	P2
3	Liushan Pu-erh tea (raw tea)	2014	9	P3
4	Liushan Pu-erh tea (raw tea)	2013	10	P4
5	Liushan Pu-erh tea (raw tea)	2012	11	P5
6	Liushan Pu-erh tea (raw tea)	2011	12	P6
7	Liushan Pu-erh tea (raw tea)	2010	13	P7
8	Liushan Pu-erh tea (raw tea)	2009	14	P8
9	Liushan Pu-erh tea (raw tea)	2007	16	P9
10	Liushan Pu-erh tea (raw tea)	2006	17	P10
11	Liushan Pu-erh tea (raw tea)	2005	18	P11
12	Liushan Pu-erh tea (raw tea)	2004	19	P12
13	Liushan Pu-erh tea (raw tea)	2003	20	P13
14	Liushan Pu-erh tea (raw tea)	2002	21	P14

**Table 2 foods-14-00840-t002:** Sensory evaluation results of Liushan Pu-erh tea (raw tea).

Sample	Appearance	Soup Color	Aroma	Taste	Leaf Base	Score ^a^
P1	Round, buds with pekoe, brownish-blackish-brown	yellower, bright	Strong fruity/floral aroma	Rich, mellow, and thick	Soft, yellowish brown	90
P2	Round, buds with pekoe, brownish-blackish-brown	yellower, bright	Moderately rich fruity/floral aroma with fragrance aroma	Mellow, thick	Soft, yellowish brown	87
P3	Round, buds with pekoe, brownish-blackish-brown	yellow, bright	Fruity/floral aroma with a slight smoky aroma	Mellow, thick	Soft, yellowish brown	88
P4	Round, buds with pekoe, brownish-blackish-brown	yellower, bright	Moderately rich fruity/floral aroma	Mellow	Soft, brownish yellow	86
P5	Round, buds with pekoe, brownish-blackish-brown	orange-yellow, bright	Strong fragrance aroma with a smoky aroma	Mellow	Soft, brownish yellow	88
P6	Round, buds with pekoe, brownish-blackish-brown	orange-yellow, bright	Strong fragrance aroma with a fruity/floral aroma	Mellow	Soft, brownish yellow	87
P7	Round, buds with pekoe, brownish-blackish-brown	orange-yellow, bright	Fragrance aroma with a slight smoky aroma	Rich, mellow, and thick	Soft, brownish yellow	88
P8	Round, buds with pekoe, brownish-blackish-brown	orange-yellow, bright	Moderately richer aroma, with a slight fruity aroma	Mellow	Soft, brownish yellow	86
P9	Round, buds with pekoe, brownish-blackish-brown	orange-yellow, bright	Moderately richer smoky aroma	Mellow, thick	Soft, brownish yellow	85
P10	Round, buds with pekoe, brownish-blackish-brown	yellowish-orange, bright	Smoky aroma, high	Mellow, thick	Soft, brownish yellow	85
P11	Round, buds with pekoe, brownish-blackish-brown	orange-red, bright	Fragrance aroma, pure	Mellow, thick	Soft, brownish yellow	85
P12	Round, buds with pekoe, brownish-blackish-brown	light orange-red, bright	Fragrance aroma, high	Mellow, thick	Soft, brownish yellow	86
P13	Round, buds with pekoe, brownish-blackish-brown, less uniform	yellow, bright	Moderately rich fragrance aroma	Mellow	Soft, brownish yellow	84
P14	Round, buds with pekoe, brownish-blackish-brown	Orange-red, moderately bright	Strong fragrance aroma and fruity/floral aroma	Mellow, thick, with a sweet aftertaste	Soft, yellowish-brown	92

^a^ Total Score = Appearance × 20% + Soup color × 10% + Aroma × 30% + Taste × 35% + Leaf base × 5%.

**Table 3 foods-14-00840-t003:** ROAV table of key aromatic substances in 14 Pu-erh tea (raw tea) samples.

No.	Compound Name	OT (ng/kg)	ROAV													
			P1	P2	P3	P4	P5	P6	P7	P8	P9	P10	P11	P12	P13	P14
1	3-Buten-2-one, 4-(2,6,6-trimethyl-1-cyclohexen-1-yl)-	0.007	20,147.91	20,583.20	19,400.35	18,535.50	25,252.84	23,872.03	30,661.60	28,457.69	21,474.77	28,412.08	28,388.07	26,644.40	20,932.07	37,231.37
2	Linalool	0.22	11,189.46	10,343.43	11,377.78	14,404.80	13,802.57	13,071.94	12,898.05	11,562.13	11,842.37	10,958.94	11,793.71	9945.34	9900.04	11,378.04
3	(E)-Damascenone	0.002	10,508.79	11,738.58	12,345.94	13,519.19	13,397.10	14,796.73	17,360.38	14,281.23	13,539.11	7838.70	12,652.16	9103.08	9968.56	10,274.00
4	4-Heptenal, (Z)-	0.0087	1886.64	15,205.95	19,964.65	-	22,993.77	-	-	18,193.11	737.82	-	21,963.93	-	16,143.70	25,079.41
5	Octanal	0.8	360.79	345.31	337.91	308.58	288.95	291.50	-	-	269.10	360.20	317.61	247.67	234.26	-
6	Furan, 2-ethyl-	2.3	-	122.12	141.97	-	143.06	-	-	117.91	-	-	118.85	80.75	85.75	111.14
7	Hexanal	4.5	-	182.36	-	193.69	257.55	1.73	181.20	189.56	-	-	5.94	-	-	-
8	Decanal	2.6	-	30.03	113.08	159.97	25.00	-	74.93	-	69.43	-	-	-	93.76	67.29
9	*β*-Myrcene	1.2	229.18	243.93	284.27	255.57	338.37	249.88	309.32	279.56	267.63	163.95	191.11	153.86	90.45	123.53
10	2-Nonenal, (E)-	0.19	214.97	163.40	166.55	189.70	170.41	241.09	202.91	180.98	355.03	143.26	395.93	182.88	177.08	374.10
11	1-Octen-3-ol	1.5	174.33	146.56	176.71	165.97	171.05	133.40	169.79	140.11	166.47	84.40	160.82	145.47	117.87	83.92
12	Heptanal	2.8	126.85	103.77	122.32	111.94	125.87	133.22	117.34	102.53	138.68	84.70	132.05	80.82	96.91	134.95
13	4-Heptenal	4.2	52.06	34.53	38.73	35.90	44.21	43.50	39.60	41.04	55.25	33.15	-	28.43	33.20	48.35
14	2-Octenal, (E)-	3	50.32	41.24	46.34	35.40	50.11	48.44	53.69	43.09	70.45	31.95	66.58	27.76	43.68	66.25
15	2,4-Heptadienal, (E,E)-	15.4	43.00	38.26	43.88	39.92	55.57	48.00	25.51	42.69	57.10	43.76	59.35	40.26	53.53	67.98
16	Naphthalene	6	27.65	34.08	34.77	30.61	39.89	43.24	41.20	34.82	50.67	34.53	34.27	28.46	27.36	42.19
17	Hexanoic acid, ethyl ester	5	26.38	13.97	20.20	16.66	16.36	20.88	23.96	10.73	19.46	7.71	22.31	2.81	5.92	3.60
18	Biphenyl	0.5	22.65	27.55	32.25	29.74	52.53	49.36	50.90	35.97	61.48	22.37	31.38	29.58	35.77	78.25
19	Styrene	3.6	18.23	12.69	13.39	18.91	24.00	21.34	22.47	19.79	23.97	18.20	22.89	20.47	13.28	21.80
20	Eucalyptol	3.1	17.05	23.48	17.29	13.04	28.28	21.56	14.92	17.40	19.95	28.39	14.09	9.88	11.53	15.37
21	*α*-Ionone	3.78–10.6	15.34	11.88	12.66	13.65	16.95	16.64	18.43	16.99	16.87	16.90	18.12	18.38	12.65	26.35
22	2-Hexenal	19.2	10.20	8.58	10.26	12.92	11.62	12.26	10.56	7.13	13.35	6.92	15.45	5.36	15.18	18.86
23	*β*-Cyclocitral	30	9.20	6.27	7.17	7.23	8.77	6.96	9.22	9.08	10.72	5.34	10.08	8.58	5.65	8.62
24	Naphthalene, 2-methyl-	3	9.13	10.51	13.13	10.67	18.86	22.37	25.28	15.50	30.91	18.27	14.31	11.42	10.96	24.68
25	2-Octanone	5	8.28	9.00	9.30	7.57	7.62	6.15	8.31	8.84	13.78	4.24	6.35	9.19	4.52	5.31
26	2-n-Butyl furan	5	7.24	5.93	7.36	6.07	7.70	6.52	9.73	6.16	9.86	3.86	5.68	3.45	1.86	4.24
27	trans-*β*-Ocimene	34	6.33	6.77	8.37	7.10	8.34	7.52	8.30	7.49	9.65	8.06	4.98	3.76	2.61	3.46
28	*α*-Phellandrene	40	5.72	4.14	4.97	3.22	4.89	4.32	4.78	4.74	4.90	2.62	3.08	2.46	1.39	2.37
29	5-Hepten-2-one, 6-methyl-	68	5.47	5.48	5.85	5.47	6.08	5.65	5.54	5.34	6.77	4.37	5.65	5.11	4.16	5.00
30	Cyclohexanone, 2,2,6-trimethyl-	100	5.42	3.81	4.37	4.79	4.33	3.69	5.13	5.77	6.17	3.17	5.76	5.15	3.37	5.00
31	Benzeneacetaldehyde	30	5.26	5.49	6.08	7.47	7.11	7.20	10.56	9.10	8.39	8.47	9.38	8.60	7.08	10.72
32	2-Decanone	8.3–41	4.88	5.88	5.20	4.55	6.60	4.58	5.56	4.38	5.98	2.42	3.79	3.96	2.07	2.84
33	*α*-Terpinene	80	4.50	4.85	5.66	4.69	5.94	5.22	6.15	5.78	6.12	3.82	4.17	3.12	1.69	3.00
34	Geraniol	6.6	4.49	7.49	7.92	10.20	10.20	7.19	11.92	7.04	6.87	4.83	6.52	4.37	6.08	5.68
35	1,3,6-Octatriene, 3,7-dimethyl-, (Z)-	34	4.44	5.27	6.45	5.64	7.18	6.01	6.59	6.30	0.68	-	4.47	3.94	1.93	3.46
36	Benzene, 1-methyl-4-(1-methylethenyl)-	85	3.97	3.98	4.08	4.05	4.83	4.60	5.40	5.17	6.67	3.00	4.81	3.05	2.28	4.29
37	Methyl salicylate	40	3.80	3.00	4.13	7.99	5.19	3.77	4.46	3.09	2.60	2.48	6.57	1.82	3.47	2.11
38	Dibenzofuran	3.3	3.59	4.83	4.87	4.35	7.63	8.78	9.41	5.86	11.60	7.15	7.24	6.41	7.52	30.95
39	terpinolene	200	2.79	2.96	3.10	2.48	3.08	2.82	3.11	3.23	3.58	2.08	2.29	1.91	0.98	1.81
40	Limonene	1000	2.53	2.31	2.51	2.24	2.81	2.61	2.83	2.75	2.99	0.02	2.32	1.81	1.32	2.15
41	2-Furanmethanol, 5-ethenyltetrahydro-*α*, *α*,5-trimethyl-, cis-	100	2.50	3.21	3.43	3.85	4.21	3.67	4.96	3.32	4.65	2.90	5.60	2.43	4.93	7.79
42	3-Octanone	21.4–21.5	1.29	1.60	2.00	1.72	2.46	2.03	2.33	2.31	2.60	2.47	2.72	1.45	1.25	2.05
43	3,5-Octadien-2-one, (E,E)-	10–150	1.23	1.00	1.11	1.23	1.17	1.55	1.49	1.28	1.43	1.30	1.45	1.17	1.26	2.10
44	2-Heptanone	140	1.19	1.16	1.40	1.20	1.32	1.21	1.40	1.41	1.91	1.08	1.57	1.11	1.19	1.55
45	trans-Linalool oxide (furanoid)	190	1.15	1.36	1.37	2.91	1.55	1.33	1.56	0.93	1.49	0.80	2.93	0.54	2.29	2.62
46	Benzaldehyde	750.89	1.09	1.02	1.13	1.01	1.27	1.26	1.54	1.55	1.73	1.43	1.47	1.38	1.21	1.84
47	2-Undecanone	5.5	0.92	1.02	1.52	1.07	1.73	1.28	1.50	1.14	1.98	1.14	1.82	1.56	0.86	1.83
48	1-Octanol	125.8	0.87	0.50	0.44	0.73	0.55	0.39	0.60	0.46	0.44	0.17	0.50	1.05	0.85	0.21
49	5,9-Undecadien-2-one, 6,10-dimethyl-, (E)-	60	0.54	0.58	0.68	0.64	0.85	0.70	0.78	0.80	0.73	0.47	0.70	0.74	0.62	1.10
50	*α*-Pinene	100	-	1.09	1.32	-	2.04	1.31	1.33	-	1.34	0.92	0.87	0.57	0.37	0.64
51	Camphene	125	-	-	-	-	-	-	-	-	1.99	2.39	1.77	1.33	-	1.59

- indicates that the aroma component is not detected in the sample.

## Data Availability

The original contributions presented in this study are included in the article/Appendix A. Further inquiries can be directed to the corresponding authors.

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
