# Peer review of "Characterization of Volatile Substances in Pu-erh Tea (Raw Tea) at Different Storage Times"

_foods, 2025, doi:10.3390/foods14050840_

Round 1
Reviewer 1 Report
Comments and Suggestions for Authors
Thank you for asking me to evaluate this manuscript titled “Characterization of Volatile Substances in Pu-erh tea (raw tea) for Different Storage Time” by Yang L. et al. The authors present their work on molecules emitted by Pu-erh tea during storage using HS-SPME-GCxGC-Q-TOF-MS and, to me, that was not original (as presented in the bibliography of the paper).
First of all, I have to say that this work was based on a very small collection of Tea samples (only 14 samples) that limits the conclusions for grouping with age or for pointing out involved molecules. It’s not clear why for example, the 3 years old Tea was not grouped with 1,2- and 4-years old samples. No explanation was given to explain that. How many replicates were done for every measurement? To me, the research design was not appropriate.
Second, the overall quality of the document does not meet Foods' requirements: the sensory evaluation is not clearly presented. The bibliography is not homogenous (some in majuscule, other not). The tables are difficult to read. Descriptions should be added to the table legends. Line 195: repetition, Line238: DecanaL, Line 207 and in the table 4: OAV or ROAV? In addition, the calculation of these values was not clear for me.
In conclusion, the collection of samples does not allow to draw conclusions without taking into account annual variations. These results do not seem to me to be sufficiently detailed to be published as they stand.
Author Response
|
Comments 1: First of all, I have to say that this work was based on a very small collection of Tea samples (only 14 samples) that limits the conclusions for grouping with age or for pointing out involved molecules. It’s not clear why for example, the 3 years old Tea was not grouped with 1,2- and 4-years old samples. No explanation was given to explain that. How many replicates were done for every measurement? To me, the research design was not appropriate.
|
|
Response 1: We sincerely express our gratitude for your evaluation. During the sampling procedure, we strictly adhered to standardized sampling protocols and collected consecutive-year Pu'er tea(raw tea) specimens. Although there are only 14 samples, the overall storage time of our Pu'er (raw tea) samples is longer compared to other studies on Pu'er (raw tea), which holds research significance. During the sampling process, we were unaware of the variation patterns of the samples. Subsequently, sensory evaluation and gas chromatography-mass spectrometry (GC-MS) tests were conducted (all tests were repeated three times). The sensory evaluation outcomes demonstrated substantial consistency with the volatile component analysis. Guided by principles of scientific rigor, sample stratification was implemented based on analytical results. The findings reveal that all specimens were stratified into three distinct subgroups according to differential aroma profiles and their respective intensity gradients. Regarding the clustering of non-consecutive storage years within identical subgroups, plausible hypotheses were posited in Section 3.6, "Discussion and Analysis." Our research sample size is relatively small, and we anticipate that future studies with additional samples will be able to supplement our conclusions.
|
|
Comments 2: Second, the overall quality of the document does not meet Foods' requirements: the sensory evaluation is not clearly presented. The bibliography is not homogenous (some in majuscule, other not). The tables are difficult to read. Descriptions should be added to the table legends. Line 195: repetition, Line238: DecanaL, Line 207 and in the table 4: OAV or ROAV? In addition, the calculation of these values was not clear for me.
|
|
Response 2:Thank you for pointing out our shortcomings. In response to the unclear presentation of sensory evaluation, we will add an aroma score radar chart to present the sensory evaluation results more intuitively. We apologize for the inconsistency of the reference format and some writing problems in the article; we will correct these errors uniformly. In the article, it should be ROAV; regarding the calculation of these values, we will provide supplements and improvements. See the attachment for specific modifications.。 |

Reviewer 2 Report
Comments and Suggestions for Authors
The manuscript deals with the assessment of volatile compounds in Pu-erh tea in relation to the time of storage. The paper has some flaws that need to be corrected:
Introduction:
Add a general description of different types of tea (black, green, white), their chemical composition and health properties. For this purpose refer to https://doi.org/10.1016/j.chemosphere.2024.143550
L33: ripe tea – it is not described
Materials and methods:
L106: Gas chromatographic…
Results and analysis – correct to Results and Discussion:
The thorough Discussion is needed. Compare your own results with other studies. Add also the results of volatiles of other types of tea.
Tables 2,3,4 are very details and should be uploaded as supplementary material. What are the units of values? Try to create general Figures or Tables from these data, reducing their size in the manuscript body. Something similar to Fig. 1
L159: CAS is not needed in Table 3
L175: add % on y axis
L207: ‘ROAV’ – full name
L219: add full names of OT and OAV in the caption
L230: ‘VIP’ – full name. Increase the dimensions of Fig. 3
L231-283: do not indicate all volatiles from each group. Focus on main and characteristic compounds determined in the highest concentration. Alternatively, remove this text, because general results were shown in point 3.5
L343: remove this heading
Conclusions:
L372-383: remove this part
Author Response
Comments 1: Introduction:
Add a general description of different types of tea (black, green, white), their chemical composition and health properties. For this purpose refer to https://doi.org/10.1016/j.chemo
sphere.2024.143550
L33: ripe tea – it is not described√
Materials and methods:
L106: Gas chromatographic…
Results and analysis – correct to Results and Discussion:
The thorough Discussion is needed. Compare your own results with other studies. Add also the results of volatiles of other types of tea.
Tables 2,3,4 are very details and should be uploaded as supplementary material. What are the units of values? Try to create general Figures or Tables from these data, reducing their size in the manuscript body. Something similar to Fig. 1
L159: CAS is not needed in Table 3√
L175: add % on y axis√
L207: ‘ROAV’ – full name√
L219: add full names of OT and OAV in the caption
L230: ‘VIP’ – full name. Increase the dimensions of Fig. 3
L231-283: do not indicate all volatiles from each group. Focus on main and characteristic compounds determined in the highest concentration. Alternatively, remove this text, because general results were shown in point 3.5
L343: remove this heading
Conclusions:
L372-383: remove this part
Response 1: We sincerely appreciate your valuable suggestions. Following your guidance, we have made several revisions. In both the introduction and subsequent analysis, we have incorporated comparisons between the sample and pu-erh tea (ripe tea) as well as other teas. We have removed the overly lengthy tables and replaced them with graphical representations of the relevant data. In the analysis section, we have added possible reasons for certain changes in compounds. Additionally, based on the content of the article, we have included several figures to facilitate a better understanding of the key points. Please refer to the attachment for the specific revisions.

Reviewer 3 Report
Comments and Suggestions for Authors
The article “Characterization of Volatile Substances in Pu-erh Tea (raw tea) for Different Storage Times” has been revised. The following observations have been made for its improvement.
A diagram explaining the difference between raw and ripe Pu-erh will greatly help us understand these differences and their commercial importance.
Likewise, a graphical abstract will help to understand the methodology and results analyzed. There is no description of how the sensory evaluation panel was chosen according to (GB/T 23776-2018) and (GB/T 22111-20 . Please be more descriptive.
A broader explanation or discussion is required of Table 3, explaining why certain components increase and then decrease, what is the reason for this cyclical behavior?
What does Figure 2 imply? What does this arrangement suggest in the different quadrants?
It is recommended that some comparisons with similar works be made.
The discussion is appropriate and broad
What would be the future work or research perspectives to be able to elucidate the best storage time and conditions?
Author Response
Comments 1: A diagram explaining the difference between raw and ripe Pu-erh will greatly help us understand these differences and their commercial importance.
Likewise, a graphical abstract will help to understand the methodology and results analyzed. There is no description of how the sensory evaluation panel was chosen according to (GB/T 23776-2018) and (GB/T 22111-20 . Please be more descriptive.
A broader explanation or discussion is required of Table 3, explaining why certain components increase and then decrease, what is the reason for this cyclical behavior?
What does Figure 2 imply? What does this arrangement suggest in the different quadrants?
It is recommended that some comparisons with similar works be made.
The discussion is appropriate and broad
What would be the future work or research perspectives to be able to elucidate the best storage time andconditions?
Response 1: Thank you for your comments. We fully agree with your suggestions. In both the introduction and subsequent analysis, we have added comparisons and connections between the samples and pu-erh tea (ripe tea) as well as other teas. We have also included explanations related to sensory evaluation. In the analysis, we have provided potential reasons for the observed changes in certain compounds. Regarding the original Figure 2 (PCA plot), we have added explanations to facilitate the understanding of the chart’s significance. Please refer to the attachment for the specific revisions. Our research results show that the samples are still in the aging transition phase, with quality improving as the storage duration increases. Based on the current findings, the sample stored for 21 years (produced in 2002) among the 14 samples demonstrates the best quality. Further investigation into samples with longer storage times remains worthwhile.

Reviewer 4 Report
Comments and Suggestions for Authors
The authors present an interesting work. However, some observations need to be performed to get a publishable category. In the attached, you can find it.

Author Response
I appreciate your valuable feedback and fully agree with your suggestions. I have incorporated an explanation of the sensory evaluation, removed the excessively long tables, and replaced them with graphical illustrations. Additionally, I have included details on the sample storage conditions. Please refer to the attachment for the specific revisions.

Round 2
Reviewer 1 Report
Comments and Suggestions for Authors
I would like to thank the authors for taking my comments into consideration. Significant changes have been made, making the document better overall. The small sample collection remains but I understand that is difficult to complete. The additions to the sensory section are clearly welcome. However, the tables and figures are still difficult to read. The list of 290 VOCs used in this study is missing and should be added in SI. I encourage therefore the authors to make these final corrections.
Author Response
Thank you very much for your suggestion. We have added more explanations to the tables and figures to improve readability. I fully agree with your suggestion and have included the supplementary file "Table S1" (list of 290 substances). The specific changes are listed in the document.

Reviewer 2 Report
Comments and Suggestions for Authors
OK
Reviewer 4 Report
Comments and Suggestions for Authors
The manuscript has been improved, however, in the document the tables are not well presented, authors need to improve it
Author Response
We sincerely appreciate your suggestions. We have made some revisions to the tables and added more explanations and analysis in the manuscript. The specific changes are outlined in the document.
